# Mechanical Characterization of Dental Prostheses Manufactured with PMMA–Graphene Composites

**DOI:** 10.3390/ma15155391

**Published:** 2022-08-05

**Authors:** Miquel Punset, Aritza Brizuela, Esteban Pérez-Pevida, Mariano Herrero-Climent, José Maria Manero, Javier Gil

**Affiliations:** 1Biomaterials, Biomechanics and Tissue Engineering Group (BBT), Department of Materials Science and Engineering, Universitat Politècnica de Catalunya (UPC), Av. Eduard Maristany 16, 08019 Barcelona, Spain; 2Barcelona Research Center in Multiscale Science and Engineering, Technical University of Catalonia (UPC), Av. Eduard Maristany 10-14, 08019 Barcelona, Spain; 3UPC Innovation and Technology Center (CIT-UPC), Technical University of Catalonia (UPC), C. Jordi Girona 3-1, 08034 Barcelona, Spain; 4Facultad de Odontología, Universidad Europea Miguel de Cervantes, C/del Padre Julio Chevalier 2., 47012 Valladolid, Spain; 5Porto Dental Institute, 4150-518 Oporto, Portugal; 6Bioengineering Institute of Technology, International University of Catalonia, Josep Trueta s/n., 08195 Barcelona, Spain

**Keywords:** dental materials, prosthetic materials, PMMA, graphene, composites, mechanical properties

## Abstract

The use of a PMMA composite with graphene is being commercialized for application as dental prostheses. The different proportions of fibers provide a wide range of colors that favors dental esthetics in prostheses. However, there are no studies that have explained the influence that graphene has on the mechanical properties. In this contribution, we studied the PMMA and PMMA material with graphene fibers (PMMA-G) in the form of discs as supplied for machining. The presence of graphene fibers has been studied by Raman spectroscopy and the Shore hardness and Vickers micro hardness were determined. Mechanical compression tests were carried out to obtain the values of maximum strength and Young’s modulus (E) and by means of pin-on-disc wear tests, the specific wear rate and the friction coefficients were determined following the established international standards. Finally, the samples were characterized by field emission scanning electron microscopy (FESEM) to characterize the graphene’s morphology inside the PMMA. The results showed the presence of graphene in PMMA and was estimated in an amount of 0.1027% by weight in G-PMMA. The Shore hardness and Vickers microhardness values did not show statistically significant differences. Differences were observed in the compression maximum strength (129.43 MPa for PMMA and 140.23 for PMMA-G) and E values (2.01 for PMMA and 2.89 GPa for PMMA-G) as well as in the lower wear rate for the G-PMMA samples (1.93 × 10^−7^ for PMMA and 1.33 × 10^−7^ mm^3^/N·m) with a *p* < 0.005. The coefficients of friction for PMMA-G decreased from 0.4032 for PMMA to 0.4001 for PMMA-G. From the results obtained, a slight content in graphene produced a significant improvement in the mechanical properties that could be observed in the prosthesis material. Therefore, we can state that the main attraction of this material for dental prosthesis is its esthetics.

## 1. Introduction

The oral cavity is one of the most aggressive human biological environments due to the complex biological and microbiological and electrolyte variation medium, which itself is affected by many type of factors such as dental plaque, bacteria oral microbiota, gastric acid reflux, and the presence of saliva as well as by variations of temperature, oxygen, and pH [1,2,3]. From the bacteria oral microbiota point of view, as the main gateway to the human body, the oral cavity shows great diversity, with more than 770 microbial species described [4,5], which can produce periodontitis and caries that are the main dental oral diseases that may lead to teeth loss [6]. In addition to the complex and variable biological conditions of the oral medium, dental materials applied in the oral cavity are exposed to extremely high mechanical loading forces [7]. In summary, these harsh factors produce the action of several degradation mechanisms including the wear, fatigue, fracture, oxidation, and corrosion of dental implant materials, among others, which can compromise the durability or lifespan of dental materials. 

Thanks to certain very useful properties such as good mechanical properties, low price, easy manipulation and repair, esthetically pleasing and high biocompatibility [8,9,10,11,12], its use has been progressively expanded and strengthened. In the dental field, the use of polymeric materials, especially PMMA, plays an important role in the manufacture of temporary oral prostheses, temporary and relieving splints, three-dimensional models, bone cements, facial prostheses, liners, obturators, temporary veneers and crowns, among other components [13,14]. Synthetic polymers have been widely used as both dental and denture materials for more than half a century, since they meet the prosthetic requirements with regard to the esthetic, mechanical properties, chemical stability, corrosion resistance, and biocompatibility to be used as a denture and dental materials [15,16,17,18,19,20,21,22]. In relation to the above-mentioned polymers, poly(methyl methacrylate) (PMMA) has been extensively used for over 80 years as a biomaterial in dental fields because it meets most of the criteria for an ideal dental and denture materials [10,22,23,24,25]. PMMA was first used as a denture material in 1937 [26]. 

However, PMMA has some weaknesses and/or drawbacks mainly related to its mechanical [8], fatigue [27], thermal [28], and biological properties [20,29]. Among the main limitations of this material for long-term use in dental applications, two fundamental aspects stand out: its mechanical properties and its propensity to bacterial infection [8]. From the point of view of the mechanical properties, the lack of hardness and toughness translates into poor wear resistance, which affects the durability of the components. Furthermore, a major drawback for their long-term is related to poor antimicrobial property, which can cause polymer deterioration as well as infection when in contact with tissues or when being implanted for restoration [8,9,20,29,30]. The PMMA denture polymer is highly susceptible to bacterial colonization by Candida albicans and other pathogens, leading to Candida-associated denture stomatitis, a complex condition with a multifactorial etiology [31,32]. This chronic inflammatory condition is common among prosthesis users, which is very difficult to treat due to incomplete disinfection of the acrylic surface and rapid microbial recolonization [33,34,35]. An improvement in the strength of denture resins and mechanical properties is necessary for the long-time behavior of this polymer. 

Several strategies have been applied in order to increase both the mechanical properties and biological behavior of PMMA polymer resins, almost all of which were based on the addition of inorganic fillers such as mesoporous silica nanoparticles (MSNs) [36,37,38,39], nano-sized graphene oxides (nGO) [16,39,40,41], titanium dioxide (TiO_2_) [41,42,43,44,45,46], and carbon nanotubes (CNTs) [47,48,49]. In this latter regard, some researchers have reported an increase in the mechanical properties coupled with the development of bactericidal capacities as a result of nGO reinforcement [41]. 

A review of the existing published scientific literature reflects the growing interest in the use/incorporation of graphene as a reinforcing phase in PMMA-based materials for dental applications [50,51,52]. Graphene is highly likely as one of the materials with the highest potential applicability in the coming decades. Since its discovery in 2004 [53], the use of graphene has expanded greatly, involving a wide variety of different fields of research, ranging from electronic and optoelectronics to biomedical in 2008. In general, graphene-based materials can be divided into four main categories: single-layer and few-layered graphene as well as graphene oxide (GO) and reduced graphene oxide (rGO) [54,55].

The dental and dental-prosthetic sectors are fields of application in continuous evolution, continuously incorporating new materials and manufacturing methods into their production processes. However, the dental field is subject to, influenced, and restricted by esthetic, biological and biocompatibility criteria, which greatly restrict the number and type of potentially usable materials including graphene.

However, the main objective of the researchers to reach a consensus on the introduction of graphene in PMMA prostheses revolves around finding the minimum concentration of graphene necessary to generate a minimal, but acceptable improvement in mechanical performance, in order to reduce the cost increase due to the high price of graphene, while reducing the risk of problems associated with esthetic criteria, the possible staining of surrounding tissues, and possible cytotoxic effects. In light of the above, the application of graphene-reinforced PMMA-based composites for the biomedical sector requires the correct balance between the desired increase in mechanical properties, compliance with esthetic criteria, and possible biological complications in the surrounding tissues.

One strategy to solve these problems is the incorporation of nano-additives (nanofibers, nanospheres, nanosheets, or nanotubes) [9,10,56,57]. Carbon-based nanomaterials, especially graphene, have been demonstrated to have a bactericide effect [9,20,58,59,60]. Graphene is a single layer of sp2 hybridized carbon atoms in a honeycomb lattice characterized by excellent properties and feasibility [61,62,63,64]. Graphene features a unique combination of several outstanding properties, some of which are related to large specific surface area (2630 m^2^ g^−1^), high thermal conductivity (5000 Wm^−1^ K^−1^), high intrinsic mobility (200,000 cm^2^ v^−1^ s^−1^), and Young’s modulus (Young’s modulus of Y ~ 1.0 TPa) [65,66,67,68,69,70].

From the point of view of biomedical applications, graphene offers several advantages such as its biocompatibility and biodegradability [8], strength [71], flexibility [72], antibacterial behavior [73], good differentiation of stem cells such as osteogenesis [74], and neurogenesis [75], among the most important. 

Many studies have been carried out on the dental applications of this PMMA composite with graphene, but there are practically no studies comparing the mechanical properties for dental prosthesis application [76,77,78,79,80,81,82,83,84,85]. The objective of this study was to determine the improvement in the PMMA when it is reinforced with graphene, especially in the compressive strength, microhardness, and wear resistance, which are the most important properties for prosthodontic applications. 

## 2. Materials and Methods

### 2.1. Materials

The samples analyzed in this study corresponded to the “G-CAM” discs (Nanographene, Murcia, Spain), which consists of a PMMA disc nano-reinforced with graphene for CAD/CAM milling (G-PMMA). The control samples were only PMMA as a control base material. Figure 1a shows the PMMA discs with graphene in different shades, from the first PMMA disc (white color) to the last one with more graphene, in which a more ochre color could be observed (Figure 1b). This color had the highest amount of graphene and is not manufactured with a higher graphene content because the color is outside the scales of dental esthetics, given that graphene is black. This is why we carried out studies with the disc with the highest graphene content, and one that was only PMMA.

The obtention of the composite (PMMA and graphene) followed the method described by Aldoari et al. [86]. The materials used were:Graphite powder with a purity higher 99.5% (Merck, Darmstadt, Germany);Hydrazine hydrate (HH, 80%) (Loba Chemi. Pvt. Ltd., Mumbai, India);Methyl methacrylate (MMA) (Acros Chemical Co., Loughborough, UK, 99%);Benzoyl peroxide (BP, BDH Chemicals Ltd., East Yorkshire, UK) was used as an initiator;Potassium permanganate (KMnO_4_, >99%) (Merck, Darmstadt, Germany);Hydrogen peroxide (H_2_O_2_, 30%) (Merck Darmstadt, Germany).

Following the Hummers and Offeman procedure [87], we obtained the graphite oxide (GO) by the oxidation of graphite. A total of 3.5 g of graphite was introduced to 98% H_2_SO_4_ (100 mL) under stirring. Then, 10 g of KMNO_4_ was introduced, maintaining the temperature below20 °C. The solution was stirred for 2 h at 35 °C. Next, the solution was poured into 500 mL of deionized H_2_O and H_2_O_2_ (ca. 20 mL of a 30% aqueous solution) in order to neutralize the excess permanganate turning the solution yellow. GO was separated using a porous glass filter and washed with HCl 10% in volume. After repeated washing of the resulting yellowish-brown cake with hot water, the GO was dried at 80 °C.

The obtained GO (400 mg) was stirred and sonicated in deionized water (20 mL) until a homogeneous yellow dispersion was obtained. HH (400 μL) was added to the solution and introduced in a microwave. The microwave (KenWood MW740, Sant Cugat del Vallés, Spain,) was operated at 900 W for 30 s cycles with stirring in order to complete the reaction (approx. 2 min) [88]. The yellow change to black color showed the completion of the reduction reaction to RGO. The RGO sheets were isolated using a centrifuge (Centurion Scientific Ltd., West Sussex, UK) working at 5000 rpm for 15 min and dried at 80 °C overnight.

RGO [0.1 (wt./wt.%)] was introduced to the monomer (MMA), stirred, and sonicated for 60 min. The initiator (BP) (5.0 wt.%) was added and dissolved in the solution. Afterward, the solution was heated to 60 °C to start the polymerization reaction using a shaking-water bath (GFL, Burgwedel, Germany). After the polymerization reaction (around 20 h), methanol was added to remove the MMA monomer. Then, the product was filtered and dried at 80 °C.

Four hundred milligrams of the dried composite of GO-PMMA was dissolved in DMF, stirred, and sonicated for 1 h. Then, the composite was placed inside a microwave oven (Kenwood MW740, Sant Cugat del Vallés, Spain) following the addition of HH (400 µL). Next, the composites were separated using a centrifuge (Centurion Scientific Ltd. West Sussex, UK) For comparison, the PMMA was prepared via a similar procedure in the absence of the RGO and GO.

The test specimens were prepared using numerical control machining techniques using a WOJIE CNC 5-axis metal milling machine model VMC 650 (WOJIE, Shandong, China). Test specimens were machined in compliance with the specifications set by international standards ASTM D2240, ASTM D695-15, and ASTM G99-17, in order to respect the dimensions specified therein.

Figure 1c shows a prosthesis fabricated with G-PMMA with the highest graphene content in which it offers very good dental esthetics. Prior to the compositional analysis and mechanical characterization, all of the samples (G-PMMA and PMMA) were subjected to a cleaning process with the following procedure:-Three washes with sodium lauryl sulfate 20% and water for 5 min in ultrasound;-Three washes in distilled water for 5 min in ultrasound;-Three washes in 96% ethanol for 5 min under ultrasound;-Drying with nitrogen N_2_ (g) for 5 min.

### 2.2. Raman Spectroscopy 

Raman spectra were obtained using a confocal Raman microscope (inViaTM Qontor confocal Raman microscope, Renishaw Inc. Waltham, MA, USA) with a 532 nm excitation laser at 100% power and a 5× objective lens. All spectra were recorded in the range 100–3500 cm^−1^ with an integration time of 45 s and 10 accumulations per spectrum. Nine measurements were performed, three measurements at three different points per sample to determine their compositional homogeneity.

### 2.3. Gel Permeation Chromatography (GPC)

The molecular weight (Mn) and polydispersity index (PDI) was determined by gel permeation chromatography (GPC) using a liquid chromatographic pump (Shimadzu, model LC-8A, Tokyo, Japan) controlled by the LC Solution software (Shimadzu, Tokyo, Japan). The polymer was dissolved end eluted in 1,1,1,3,3,3-hexafluoroisopropanol (HFIP) containing CF3COONa (0.05 M). The flow rate was 1 mL/min, the injected volume of 20 mm, and the sample concentration of 6 mg/mL. 

A PL HFIPgel column (Agilent Technologies Deutschland GmbH, Boblingen, Germany) and a refractive index detector (Shimadzu, model RID-20A, Tokyo, Japan) were employed. The number and weight average molecular weights were determined using the PMMA standards purchased from Sigma Aldrich, St. Louis, MI, USA.

### 2.4. Shore Hardness (HSD)

The shore surface hardness is determined from the elastic reaction of a material when a rounded diamond-tipped indenter is dropped on it. The hardness depends on the amount of energy absorbed by the test material during impact. For this test, the specifications indicated in UNE-EN ISO 868:2003-Plastics and ebonite were used. The hardness Shore-D (HSD) is then proportional to the hammer rebound height, according to Expression (1) [89] where K is the proportionality factor; *H*_1_ is the hammer drop height; *H*_2_ is the hammer rebound height.
(1)$$\mathrm{HSD}=K\cdot \frac{H_{1}}{H_{2}}$$

Determination of the indentation hardness was undertaken by means of a hardness tester (Shore hardness) [1] using a BAXLO equipment model SHORE TYPE D-U (Barcelona, Spain). In this case, a type D Shore indenter was used, applying a load of 4 kg for 15 s, respecting a distance of 12 mm with respect to the external perimeter of the sample and between consecutive measuring points, as specified in the ASTM D2240 test standard. A total of 10 measurements were taken per sample. The BAXLO hardness tester used in this study has a sensitivity of 1 Shore-D unit, a measuring capacity ranging from 1 to 100 Shore units and a measuring standard of 60 Shore-D units. 

A total of five Shore-D hardness measurements were carried out on each of the 15 samples analyzed, observing a distance of 12 mm with respect to the external perimeter of the sample and between consecutive measuring points, as specified in the ASTM D2240 test standard. 

### 2.5. Hardness by Vickers Indentation

The determination of hardness was analyzed by using an Izasa Durascan G5 microhardness tester (Izasa Scientific, Madrid, Spain) equipped with a Vickers indenter, which consists of a diamond pyramid with a base angle of 136°. The hardness measurements were carried out under a constant load of 300 g applied for 15 s, making a total of six measurements for each of the two materials studied.

The Vickers hardness number (HVN) in GPa can be expressed following Equation (2). HVN as a function of the applied load (P) in N and the average of the diagonals of the indentation (*d*) in μm [87]: The constant value, 1854.4, was obtained from the calculation of the contact area.
(2)$$\mathrm{HVN}=1854.4\;P/d^{2}$$

### 2.6. Compression Test

Both the elastic modulus and the maximum fracture toughness were determined by means of a compression test carried out under the specifications indicated in ASTM D695-15 “Standard Test Method for Compressive Properties of Rigid Plastics” [3]. For this purpose, a Bionix 358 (MTS, Eden Prairie, MN, USA) servo-hydraulic mechanical testing machine was used at a constant speed of 200 mm/min. 

The maximum stress was analyzed using five specimens of G-PMMA material and five control specimens of PMMA, which had a diameter of 12 mm and a height of 25 m, as indicated in the standard used. The Young’s modulus was determined using five specimens of the G-PMMA and PMMA materials with a diameter of 12 mm and a height of 50 mm, as indicated in the standard.

### 2.7. Surface Wear (“Pin-on-Disc Test”)

The ball-on-disc test consists of evaluating the wear resistance of a material by sliding a metal ball over a surface with a constant load, obtaining values for the coefficient of friction and wear rate. In this case, a CSEM tribometer (CSM Instruments, Peseux, Switzerland) was used following the specifications indicated in ASTM G99-17 “Standard test method for wear testing with a pin-on-disk apparatus” [2], using a stainless steel ball of 6 mm in diameter with a normal load of 20 N applied. The equipment used can be observed in Figure 2. The amount of material loss was evaluated by measuring the cross-sectional area of the wear tracks using a Taylor-Hobson profilometer (Talysurf Plus Roughness tester).

The friction coefficient values were measured in real-time throughout the whole experiment. The friction coefficient values were measured in real-time throughout the whole experiment. The Ws (specific wear rate) value was determined by measuring the volume loss of material in the wear scar channel after the tests were performed, according to Expression (3) [88], where V is the total wear volume in wear channel after the sliding distance L under the normal contact load W.
(3)Ws=V/(W·L)

Tests were carried out with the immersion lubrication of the samples in Hank’s physiological medium at 37 °C during a total travel distance of 500 m. All tests were performed at a constant linear pin speed of 10 cm/s with a turning radius of 4 mm. In all of the tests, a stainless steel spherical counter body pin with a diameter of 6 ± 0.001 mm, hardness of 62 HRC, and average surface roughness (Ra) of 0.02 µm was used as a friction pair. This counter body has a Young’s modulus of 200 GPa and a Poisson’s ratio of 0.27. Three specimens were realized per each material.

### 2.8. Field Emission Scanning Electron Microscopy (FESEM)

A field emission scanning electron microscope FSEM model “JSM-7001F Scanning Microscope” (Jeol, Tokyo, Japan) using a potential of 5 KV and an approximate working distance of between 9 and 11 mm was used for the measurements. This was equipped with an EDS (energy-dispersive X-ray spectroscopy, OXFORD model Xmax20, Oxford, UK), which allows for the identification of the chemical composition by means of the acquisition of the characteristic X-ray emission of each chemical element. In order to ensure the correct electrical conductivity of the PMMA surface with graphene, the sample was previously coated with a nanometric platinum/palladium (Pt–Pd) film by PVD-sputtering (Wilmington, MA, USA).

### 2.9. Statistical Analysis

The measurements were analyzed with the Stata 14 package (StataCorp^®^, College Station, San Antonio, TX, USA). Statistical values such as the averages and standard deviations were determined. 

## 3. Results

Molecular weight (Mn) and polydispersity index (PDI) were determined by GPC using a liquid chromatographic pump (Shimadzu, model LC-8A, Tokyo, Japan) controlled by the LC Solution software (Shimadzu, Tokyo, Japan). The polymer was dissolved end eluted in 1,1,1,3,3,3-hexafluoroisopropanol (HFIP) containing CF3COONa (0.05 M). All parameters were measured in duplicate. The results obtained through the GPC analysis are summarized in Table 1, expressing the results in the form of a range of values. From the molecular mass distributions, several average parameters were obtained such as the average molecular mass in number (Mn), the average molecular mass in weight (Mw), the polydispersity (PDI), and the average molecular mass (Mz).

The RAMAN fingerprint of the PMMA and G-PMMA materials can be observed in Figure 3a,b, respectively. Regarding the G-PMMA material, the graph presented in Figure 3b shows three Raman spectra corresponding to the three different surface points of the analysis of the same sample. In the literature, the characteristic peaks for graphene are identified by RAMAN around 1582–1587° and above 2688–2790° [6,19]. The analyses performed on the PMMA sample with graphene showed two peaks corresponding to these positions: a well-defined peak around 1585° and a softer peak around 2689° (peaks marked with arrows in Figure 3b). These two peaks appeared in the analyzed areas of the sample. The content of graphene estimated was 0.1027 wt%.

The first step of the mechanical characterization of the materials under study consisted of the determination of hardness, which was carried out using two different methods (Shore-A and HVN), the results of which are shown graphically in Figure 4. 

The results of the determination of the Shore-D HSD hardness were for PMMA 81 ± 4 Shore units and for G-PMMA 81 ± 4 Shore-D units. The statistical analysis of the hardness results did not show the existence of statistically significant differences between the two samples analyzed. 

The Vickers micro hardness results were 23.6 ±1.1 HVN for PMMA and for G-PMMA 24.6 ± 0.7. Again, no statistically significant differences were observed in this measure of hardness. Non-statistically significant differences in both the Shore-A and Vickers hardness values were observed between the two groups of samples (*p* > 0.05). The results of hardness can be observed in Table 2. 

From the compression tests for the PMMA and G-PMMA were obtained the results of the elastic modulus and maximum strength to fracture. Both materials evaluated showed the typical fracture behavior of markedly brittle, without the presence of clear signs of plastic deformation regardless of being reinforced with graphene. 

These results can be observed in Table 3. The graphical representation of the parameters determined by the mechanical compression tests are shown in Figure 5 in the form of bar graphs. Statistically significant differences in both the maximum compressive strength and elastic modulus were observed between the two groups of samples (*p* < 0.05). 

From the pin-on-disc wear test, the results corresponding to the weight loss and friction coefficient were obtained. Table 4 shows the SWR (specific wear rate) values for each material studied in relation to the load and distance travelled. Table 4 shows the friction coefficients obtained (µ_m_) for each material analyzed.

The graphical representation of the parameters determined by wear tests are shown in Figure 6 in the form of bar graphs. Non-statistically significant differences in friction coefficient were observed between the two groups of samples (*p* > 0.05).

The analysis of the results presented in Table 4 seems to indicate a lower specific wear rate of the G-PMMA material with respect to PMMA. Statistically significant differences in wear rate were observed between the two groups of samples (*p* < 0.005). 

The analysis of the results in Table 4 showed no differences in the friction coefficient between the two materials analyzed. The variation in the friction coefficient (μ) as a function of the sliding distance during the wear tests for the PMMA and G-PMMA materials is presented in Figure 7, respectively, where a similar behavior of the two materials analyzed can be seen as not showing statistically significant differences (*p* > 0.005).

A detailed analysis of the graphs shown in Figure 7 revealed that all of the tests experienced a running-in period followed by a gradual stabilization in the friction coefficient value, regardless of the type of material tested. The fluctuations observed for all of the friction coefficients curves were due to the formation of debris particles in the contact and their subsequent evacuation outside the wear channel.

The surface observation of the sample by SEM microscopy showed roughness due to the machining of the sample. The analysis of point composition (EDS) and the distribution of chemical element “mapping” on the surface made it possible to detect some graphene nanosheets such as the one shown in Figure 8. However, it was not possible to see a large number of them and those observed did not appear to follow preferential directions.

## 4. Discussion

The use of graphene as a reinforcement in composites has a high efficiency in mechanical properties, with weight concentrations of 2.5% of the composite material being used in different applications [76]. These contents are not suitable for the application of prosthetic materials that have to have good esthetics. When concentrations exceed 0.35% of graphene in PMMA, it becomes a very dark color, which is not acceptable for good dental esthetics [90]. The concentration of graphene in our study was analyzed through the Raman spectra and was quantified and confirmed by the phase rule as 0.027% by weight. The form observed by FESEM was in the form of nanosheets and corresponded to carbon; in no case was carbon oxide visible. As shown in Figure 8, graphene was in the form of small flakes with an average size of 100–200 nm.

As observed, the increase in micro hardness due to the incorporation of graphene was very small, from 23.6 for PMMA to 24.6 HVN for PMMA-G, where no statistical difference significance was observed. The modulus of elasticity values converted the material with a higher modulus of elasticity from 2.01 for PMMA to 2.89 GPa for PMMA-G. These differences had a statistical significance of *p* < 0.05. The mechanical resistance to compression also caused a slight increase, making it more resistant from 129.43 to 140.23 MPa with a statistical difference significance of *p* < 0.05. All of these mechanical properties led to an increase in the wear resistance, with wear rates for PMMA from 1.93 × 10^−7^ to 1.33 × 10^−7^ g/Nm for PMMA-G. The differences in wear rate had a statistical significance difference with *p* < 0.05 and a decrease in the friction coefficient from 0.4032 to 0.4001 due to the addition of graphene, which had a statistical significance difference with *p* < 0.05. These properties are very important for prosthetic materials due to the chewing and occlusal function of the teeth, causing wear between materials. 

The effect of graphene nanoparticles released into the oral cavity should be studied. It is important to know the effect of these carbon nanoparticles due to their excellent hemocompatibility, and therefore will most likely enter the blood circuit [91,92]. However, PMMA debris has been shown to be bioinert and is either phagocytosed by macrophages or a fibrotic capsule is formed around the PMMA particle when it is larger than 20 µm [[90], [92][93], [94], [95], [96], [97], [98]].

Alamgir et al. [99] observed a greater resistance to deformation of the PMMA–graphene composite with a higher value of elastic modulus than PMMA itself. In addition, the work of Khan et al. [100] found that increasing the flexion by three points with a concentration of 0.048% graphene increased the flexural deflection to 87 MPa compared to 66 MPa for PMMA. It should be noted that the addition of graphene increases the glass transition temperature and therefore avoids embrittlement in the body temperature range while maintaining good mechanical strength. When more than 1% of graphene (nanosheets) is added, the material produces an 80% increase in the modulus of elasticity and a 20% increase in the mechanical strength of PMMA, as demonstrated by Ramanathan et al. [101,102].

In addition to the mechanical properties, future studies with this composite should be evaluated including the plaque index (PI), bleeding on probing (BOP), and abnormal probing pocket depth (PPD), which Tecco et al. [103] proved to be significant risk indicators for secondary implant failure due to peri-implantitis in patients rehabilitated with cemented prosthesis. Another limitation of the study is the clinical application of the material as a prosthetic material and comparison with other materials such as zirconia. Cappare et al. [104] conducted a study comparing prosthetic materials in 50 patients with 222 dental implants placed with different prostheses: metal–acrylic and monolithic zirconia with no influence of the material on the failure rate or complications between them. A study following the protocol of Cappare et al. would be necessary to compare with the PMMA–graphene. On the other hand, another aspect that should be studied with this PMMA–graphene composite is the use of new technologies in the provisional and definitive prosthetic phase using digital technologies. Digital technology has allowed for a clear reduction in working times and costs and has allowed for the reduction in stress for patients who undergo invasive and extensive treatments to recover esthetics and function, and for clinicians who must manage complex cases with fewer appointments possible. The use of this new composite in digital dentistry is to be determined [105,106,107,108].

Regarding the potential clinical importance of the use of G-PMMA as a dental material, two main aspects should be highlighted: the improvement of the properties and the corresponding increase in the durability of the prostheses and dentures as well as the incorporation of bactericidal capabilities to the devices, with the corresponding increase in the useful life of the components as well as in the quality of life and the oral health of the patients. 

## 5. Conclusions

The composite was characterized by FESEM and the graphene content by Raman spectroscopy. The sizes of the nanosheets ranged from 100 to 200 nm. The graphene content in the composite was found to be 0.027 wt%. This addition in PMMA produces a composite that can be used as a dental prosthesis. The addition of graphene nanosheets has been shown to result in increases in the elastic modulus, compressive strength, and a significant reduction in the specific wear rate. These properties considerably improve the performance of the biocomposite in its application as a dental prosthesis material.

Although a slightly statistically significant improvement in the mechanical properties and wear resistance was achieved with the addition of 0.027% by weight of graphene, future studies will be required to determine the optimum reinforcement percentage range. In light of the above, the application of graphene-reinforced PMMA-based composites for the biomedical sector will require the correct balance between the desired increase in mechanical properties, compliance with the esthetic criteria, and possible biological complications in the surrounding tissues.

Future studies will be necessary in order to determine the influence of graphene morphology such as nanosphere morphologies, nanosheets... on the mechanical properties. Biocompatibility studies should be carried out, especially on the particles released by wear in soft tissues and on the possibility of carbon nanoparticles entering the bloodstream. Finally, it would be necessary to determine the fatigue tests of this composite.

## Figures and Tables

**Figure 1 materials-15-05391-f001:**
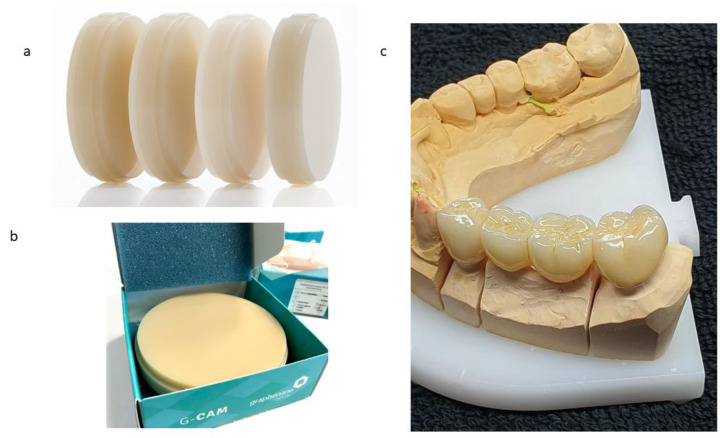
(**a**). Color range of PMMA–graphene offered by the manufacturer. (**b**) Samples studied as G-PMMA. (**c**) Dental restoration with G-PMMA with excellent esthetic results.

**Figure 2 materials-15-05391-f002:**
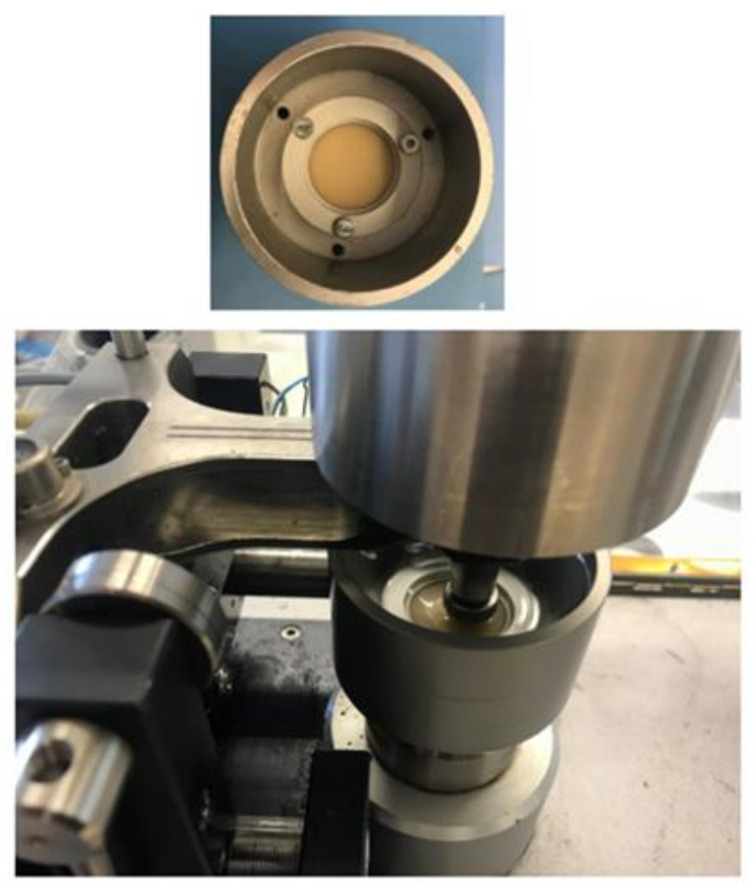
The pin-on-disc test setup used for the wear resistance study.

**Figure 3 materials-15-05391-f003:**
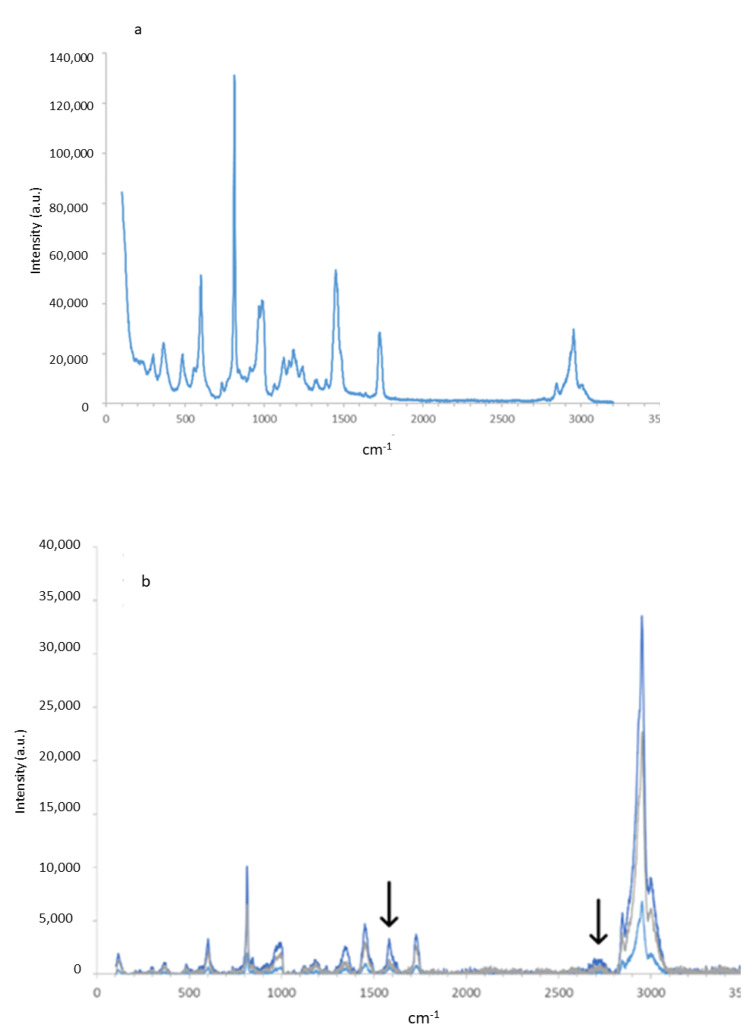
The Raman spectra of the PMMA (**a**) and G-PMMA (**b**) materials tested.

**Figure 4 materials-15-05391-f004:**
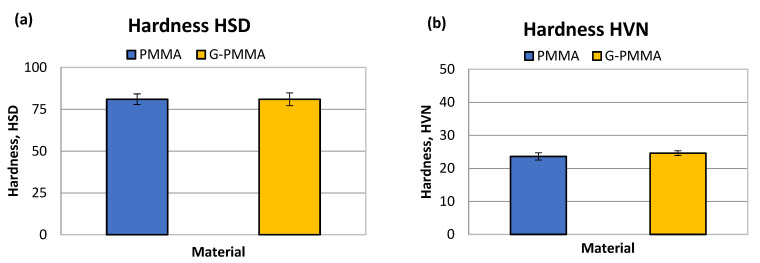
The hardness HSD (**a**) and HVN (**b**) obtained by mechanical testing of the PMMA and G-PMMA materials.

**Figure 5 materials-15-05391-f005:**
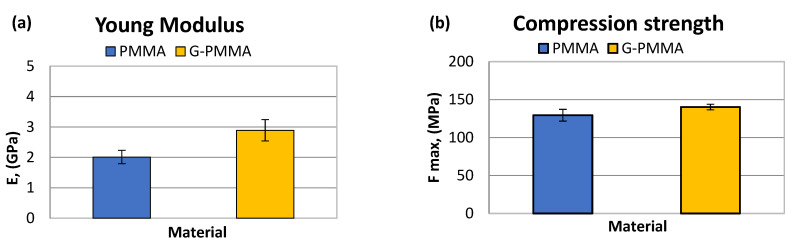
The Young’s modulus (**a**) and compression strength (**b**) obtained by mechanical testing of the PMMA and G-PMMA materials.

**Figure 6 materials-15-05391-f006:**
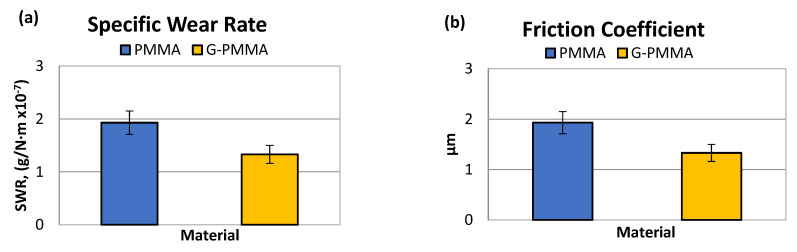
The wear rate (**a**) and friction coefficient (**b**) obtained by the pin-on-disk tests of the PMMA and G-PMMA materials.

**Figure 7 materials-15-05391-f007:**
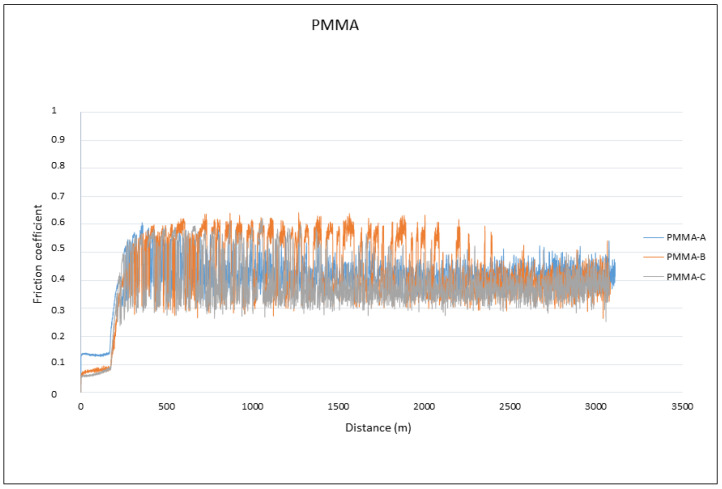
The friction coefficients against the travelled distance for each material for PMMA and PMMA-graphene.

**Figure 8 materials-15-05391-f008:**
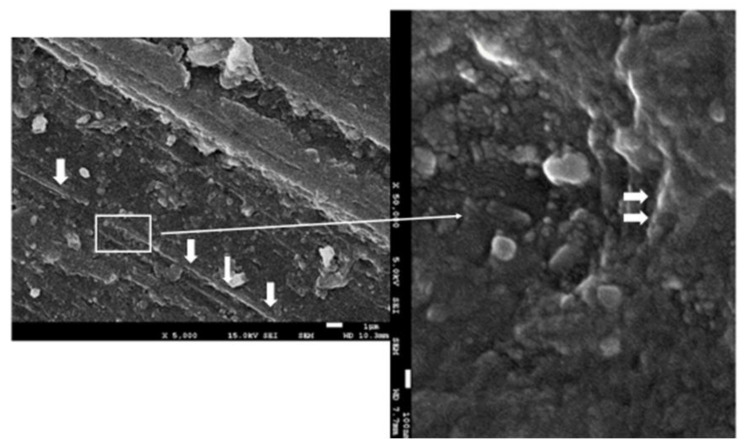
The FESEM images of the G-PMMA surface. Nanosheets of graphene can be observed in the PMMA matrix. The thinner arrow indicates the zone (rectangle) which more magnification. The thicker arrow indicates the graphene component.

**Table 1 materials-15-05391-t001:** Values obtained by GPC on the PMMA base.

	Mn [g·mol^−1^]	Mw [g·mol^−1^]	Mz [g·mol^−1^]	Mz [g·mol^−1^]	PDI [a.u]
PMMA	51,029–54,175	104,175–107,231	240,674–251,413	663,145–698,799	1.979–2.041

**Table 2 materials-15-05391-t002:** The Shore-D and Vickers hardness values for the PMMA and G-PMMA samples.

	HSD	HVN
PMMA	81 ± 3.8	23.6 ± 1.1
G-PMMA	81 ± 3.2	24.6 ± 0.7

**Table 3 materials-15-05391-t003:** The elastic modulus and maximum compression strength for the PMMA and G-PMMA samples.

	E (GPa)	Fmax (MPa)
PMMA	2.01 ± 0.22	129.43 ± 7.75
G-PMMA	2.89 ± 0.35	140.23 ± 3.69

**Table 4 materials-15-05391-t004:** The specific wear rate and friction coefficient for the PMMA and G-PMMA materials.

	SWR (g/N·m)	µm (MPa)
PMMA	1.93 × 10^−7^ ± 0.22 × 10^−7^	0.4032 ± 0.0251
G-PMMA	1.33 × 10^−7^ ± 0.17 × 10^−7^	0.4001 ± 0.0109

## Data Availability

Not applicable.

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
