# Peer review of "Mechanical Characterization of Dental Prostheses Manufactured with PMMA–Graphene Composites"

_materials, 2022, doi:10.3390/ma15155391_

Round 1
Reviewer 1 Report
Formulas for mechanical properties tests are missing.
Add P-value in both summary and results in main text.
Add tables of mechanical test values and not just graphs.
If they followed ISO for sample preparation include them as only a few are mentioned.
Add a paragraph in the discussion mentioning the clinical significance of the material tested.
Question: do the authors think that fatigue test of the specimens before testing would be important and why didn't they do it?
Author Response
Dear Reviewer,
Thanks for taking the time to review our manuscript and suggest to us to improve our work by providing a lot more detail. We have done so, and we are now submitting a manuscript that not only addresses the points that you specifically raised but also many others that we have considered in order to deliver what we think is a much improved version of our work. This version includes more paragraphs, English grammar revisions in all main sections and new references. Thanks a lot. We are looking forward to your comments.
Sincerely,
Javier Gil
REVIEWER N1
Open Review
(x) I would not like to sign my review report
( ) I would like to sign my review report
English language and style
( ) Extensive editing of English language and style required
(x) Moderate English changes required
( ) English language and style are fine/minor spell check required
( ) I don't feel qualified to judge about the English language and style
Yes |
Can be improved |
Must be improved |
Not applicable |
|
Does the introduction provide sufficient background and include all relevant references? |
( ) |
(x) |
( ) |
( ) |
Are all the cited references relevant to the research? |
( ) |
( ) |
(x) |
( ) |
Is the research design appropriate? |
( ) |
( ) |
(x) |
( ) |
Are the methods adequately described? |
( ) |
(x) |
( ) |
( ) |
Are the results clearly presented? |
( ) |
( ) |
(x) |
( ) |
Are the conclusions supported by the results? |
( ) |
(x) |
( ) |
( ) |
Comments and Suggestions for Authors
In this manuscript, the author investigated the mechanical properties of dental prostheses made of PMMA-graphene composites. From the results obtained, a slight content of graphene produces an significant improvement of the mechanical properties can be observed in the prosthesis material. However, there are some issues need to be addressed. Therefore, a major revision of this manuscript is recommended.
(1) The graphene is nanosheet, rather than fibers or nanofibers.
DONE: The text has been rechecked and the error rectified. The text had been revised accordingly to this comment.
(2) In the abstract, the “129,43” should be changed to “129.43”
DONE: The abstract has been rechecked and the error rectified. The text had been revised accordingly to this comment.
(3) In the section of “2.1 Materials”, “a PMMA biopolymer”, the PMMA is not a biopolymer, but a petroleum based polymer.
DONE: The text had been revised accordingly to this comment and the definition of PMMA as a biopolymer has been deleted.
(4) How much graphene was added into the PMMA?
DONE: The concentration of graphene in our study has been analyzed in Raman spectra and has been quantified and confirmed by the phase rule as 0.027% by weight
(5) What is the procedure for the preparation of composites discs?
A new paragraph has been introduced about this aspect in materials and methods.
(6) In Figure 1, the color of graphene is black, the obtained composites discs should be grey, why does the obtained composites discs is light yellow? Is there any other additive?
DONE by Justification: The black color of graphene causes a darkening of the PMMA as the percentage of reinforcement increases, a reason that limits the maximum concentration of use for aesthetic criteria. The colouring of the discs is adjusted with the use of other colouring additives that have no effect on the mechanical properties.
(7) The resolution of the Raman spectra should be improved. Why there are three curves in the G-PMMA Raman spectra?
DONE: The three curves presented in G-PMMA Raman spectra correspond to a three different analyses of the same sample. The article has been revised and the following paragraph has been added:
“The RAMAN fingerprint of PMMA and G-PMMA materials are observed in Figure 3a and Figure 3b, respectively. Regarding G-PMMA material, the graph presented in the Fig-ure 3b shows three Raman spectra corresponding to three different surface points of anal-ysis of the same sample.”
(8) There are more than one composites discs, however, there is only one mechanical properties (Table 1 and 2) for the composites, the mechanical properties with different graphene content should be provided.
DONE by justification: Only two materials have been characterized in this study, and more specifically to PMMA and G-PMMA materials.
(9) The format of the references should be revised, and the DOI number should be provide, according to the journal’s requirement.
DONE: The format of the bibliographical references has been revised. The DOI number has been added in all the references.
Submission Date
03 June 2022
Date of this review
08 Jun 2022 07:28:55
Reviewer 2 Report
In this manuscript, the author investigated the mechanical properties of dental prostheses made of PMMA-graphene composites. From the results obtained, a slight content of graphene produces an significant improvement of the mechanical properties can be observed in the prosthesis material. However, there are some issues need to be addressed. Therefore, a major revision of this manuscript is recommended.
(1) The graphene is nanosheet, rather than fibers or nanofibers.
(2) In the abstract, the “129,43” should be changed to “129.43”
(3) In the section of “2.1 Materials”, “a PMMA biopolymer”, the PMMA is not a biopolymer, but a petroleum based polymer.
(4) What is the Mn and PDI of the used PMMA?
(5) How much graphene was added into the PMMA?
(6) What is the procedure for the preparation of composites discs?
(7) In Figure 1, the color of graphene is black, the obtained composites discs should be grey, why does the obtained composites discs is light yellow? Is there any other additive?
(8) The resolution of the Raman spectra should be improved. Why there are three curves in the G-PMMA Raman spectra?
(9) There are more than one composites discs, however, there is only one mechanical properties (Table 1 and 2) for the composites, the mechanical properties with different graphene content should be provided.
(10) The format of the references should be revised, and the DOI number should be provide, according to the journal’s requirement.
Author Response
Dear Reviewer,
Thanks for taking the time to review our manuscript and suggest to us to improve our work by providing a lot more detail. We have done so, and we are now submitting a manuscript that not only addresses the points that you specifically raised but also many others that we have considered in order to deliver what we think is a much improved version of our work. This version includes more paragraphs, English grammar revisions in all main sections and new references. Thanks a lot. We are looking forward to your comments.
Sincerely,
Javier Gil
REVIEWER N2:
Open Review
(x) I would not like to sign my review report
( ) I would like to sign my review report
English language and style
( ) Extensive editing of English language and style required
( ) Moderate English changes required
(x) English language and style are fine/minor spell check required
( ) I don't feel qualified to judge about the English language and style
Yes |
Can be improved |
Must be improved |
Not applicable |
|
Does the introduction provide sufficient background and include all relevant references? |
(x) |
( ) |
( ) |
( ) |
Are all the cited references relevant to the research? |
(x) |
( ) |
( ) |
( ) |
Is the research design appropriate? |
(x) |
( ) |
( ) |
( ) |
Are the methods adequately described? |
(x) |
( ) |
( ) |
( ) |
Are the results clearly presented? |
(x) |
( ) |
( ) |
( ) |
Are the conclusions supported by the results? |
(x) |
( ) |
( ) |
( ) |
Comments and Suggestions for Authors
- Formulas for mechanical properties tests are missing.
DONE: According to the reviewer's commentary, Formulas for mechanical properties has been introduced.
- Add P-value in both summary and results in main text.
DONE: According to the reviewer's coemmentary, the propability values have been added both in the abstract and in the results section.
- Add tables of mechanical test values and not just graphs.
DONE: The article has been revised and the graphs and tables of results have been added.
- If they followed ISO for sample preparation include them as only a few are mentioned.
DONE: The section on materials and methods has been revised and the following paragraph has been added.
“The test specimens were prepared using numerical control machining techniques, using a WOJIE CNC 5-axis metal milling machine model VMC 650 (WOJIE, Shan dong, China). Test specimens were machined in compliance with the specifications set by international standards ASTM D2240, ASTM D695-15 and ASTM G99-17, in order to respect the dimensions specified therein.
- Add a paragraph in the discussion mentioning the clinical significance of the material tested.
DONE: The section of discussion has been revised and the following paragraph has been added.
“Regarding the potential clinical importance of the use of G-PMMA as a dental material, two main aspects should be highlighted: the improvement of the properties and the corresponding increase in the durability of prostheses and dentures, as well as the incorporation of bactericidal capabilities to the devices, with the corresponding increase in the useful life of the components as well as in the quality of life and oral health of the patients.”
- Question: do the authors think that fatigue test of the specimens before testing would be important and why didn't they do it?
DONE by justification: In response to the reviewer's question, we believe that fatigue testing will be of great importance in future work on this type of material, but it has not been the subject of study in this article. Fatigue tests are economically expensive and time consuming, which is why they were initially discarded for this study.
Submission Date
03 June 2022
Date of this review
24 Jun 2022 08:57:21
Reviewer 3 Report
The reviewed manuscript presents the results of a comparative study of two materials used in dental prosthetics. PMMA and PMMA modified with graphene were tested.
On the basis of the presented research methodology, it is impossible to clearly define what materials were tested. The authors did not describe both the raw materials and the method of obtaining test samples. In my opinion, in an article in Materials, this is a priority and necessary.
It is known that PMMA in its pure form is not used in prosthetics, so there must be other additives in the tested system. Additionally, how was the filler dispersed in the polymer / monomer (depending on the manufacturing technique of the prosthetic materials)?
Without providing a detailed methodology for obtaining materials, the article becomes useless for the reader because he is not able to reproduce the received material. The article in this form is of no importance for the development of science.
The authors performed the research reliably and, importantly, the results were provided with an appropriate statistical analysis. However, the scientific novelty of the presented materials is slight. There are many scientific reports on acrylate-based dental materials modified with graphene. Note that the main conclusions drawn from the research presented can be found on the material manufacturer's website (www.graphenanodental.com).
In my opinion, the article is only a research report of two materials obtained by a commercial producer, which has the characteristics of an advertisement rather than a scientific article.
Additionally, the authors' use of the term "biomaterial" for PMMA and graphene is puzzling. These materials are neither biodegradable nor derived from materials of plant or animal origin. These materials can at best be called biocompatible material.
Summing up, in my opinion, the article is not suitable for publication in a journal dealing with research of materials, and the scientific level of the presented research is too low for the reputation of the Materials journal.
Author Response
Dear Reviewer,
Thanks for taking the time to review our manuscript and suggest to us to improve our work by providing a lot more detail. We have done so, and we are now submitting a manuscript that not only addresses the points that you specifically raised but also many others that we have considered in order to deliver what we think is a much improved version of our work. This version includes more paragraphs, English grammar revisions in all main sections and new references. Thanks a lot. We are looking forward to your comments.
Sincerely,
Javier Gil
REVIEWER N3
(x) I would not like to sign my review report
( ) I would like to sign my review report
English language and style
( ) Extensive editing of English language and style required
( ) Moderate English changes required
( ) English language and style are fine/minor spell check required
(x) I don't feel qualified to judge about the English language and style
Yes |
Can be improved |
Must be improved |
Not applicable |
|
Does the introduction provide sufficient background and include all relevant references? |
( ) |
( ) |
(x) |
( ) |
Are all the cited references relevant to the research? |
(x) |
( ) |
( ) |
( ) |
Is the research design appropriate? |
( ) |
( ) |
(x) |
( ) |
Are the methods adequately described? |
( ) |
( ) |
( ) |
(x) |
Are the results clearly presented? |
( ) |
( ) |
( ) |
(x) |
Are the conclusions supported by the results? |
( ) |
( ) |
( ) |
(x) |
Comments and Suggestions for Authors
The reviewed manuscript presents the results of a comparative study of two materials used in dental prosthetics. PMMA and PMMA modified with graphene were tested.
On the basis of the presented research methodology, it is impossible to clearly define what materials were tested.
- The authors did not describe both the raw materials and the method of obtaining test samples. In my opinion, in an article in Materials, this is a priority and necessary.
DONE: The section on materials and methods has been revised and the following paragraph has been added.
“The test specimens were prepared using numerical control machining techniques, using a WOJIE CNC 5-axis metal milling machine model VMC 650 (WOJIE, Shan dong, China). Test specimens were machined in compliance with the specifications set by international standards ASTM D2240, ASTM D695-15 and ASTM G99-17, in order to respect the dimensions specified therein.
- It is known that PMMA in its pure form is not used in prosthetics, so there must be other additives in the tested system. Additionally, how was the filler dispersed in the polymer / monomer (depending on the manufacturing technique of the prosthetic materials)?
In agreement with the reviewer, the methodology used to obtain the nanocomposite has been introduced in order to better interpret the results. We would like to thank the reviewer for this suggestion, which has significantly improved the quality of the paper.
- Without providing a detailed methodology for obtaining materials, the article becomes useless for the reader because he is not able to reproduce the received material. The article in this form is of no importance for the development of science.
The methodology has been introduced in detail in agreement with the reviewer.
- The authors performed the research reliably and, importantly, the results were provided with an appropriate statistical analysis. However, the scientific novelty of the presented materials is slight.
In our opinion, the contribution describes in an original way the most important mechanical properties for prostheses. There are practically no studies of the wear of the PMMA-graphene prosthesis material, nor are there studies of the compressive strength, which are the forces exerted in human mastication. We also believe that the electron microscopy study of the graphene fragments is noteworthy.
- There are many scientific reports on acrylate-based dental materials modified with graphene. Note that the main conclusions drawn from the research presented can be found on the material manufacturer's website (graphenanodental.com).
In my opinion, the article is only a research report of two materials obtained by a commercial producer, which has the characteristics of an advertisement rather than a scientific article.
We believe that the reviewer's comment is not fair as the website does not provide values for hardness types, friction coefficients, wear resistance and high resolution electron microscopy images. The authors believe that it is a contribution that contributes to and complements many of the studies carried out on this composite. The other reviewers have highlighted this fact. I think that with all the new contributions we have made in the review, the reviewer may change his opinion. I think as a member of the editorial board of Materials that with the modifications thanks to the comments especially of this reviewer it has improved so that the opinion about this work can change. Thank you very much for your understanding and work.
- Additionally, the authors' use of the term "biomaterial" for PMMA and graphene is puzzling.
These materials are neither biodegradable nor derived from materials of plant or animal origin.
These materials can at best be called biocompatible material.
DONE: The text had been revised accordingly to this comment and the definition of PMMA as a biopolymer has been deleted and vhenged to dental biocompatible material.
Summing up, in my opinion, the article is not suitable for publication in a journal dealing with research of materials, and the scientific level of the presented research is too low for the reputation of the Materials journal.
Submission Date
03 June 2022
Date of this review
14 Jun 2022 20:04:52
Round 2
Reviewer 2 Report
Though the author made a revision of this manuscript, yet more serious issues were appeared. The revised manuscript was also in bad organizing, , which is too primary to be a scientific paper at this stage. Therefore, a Rejection of this manuscript is recommended.
(1) The question “What is the Mn and PDI of the used PMMA?” present in the first round of review was not answered.
(2) The question “What is the procedure for the preparation of composites discs?” present in the first round of review was also not answered.
(3) As we all know the PMMA should be transparent and colorless, why does the PMMA obtained here is white color and opaque? Some white inorganic filler mus t be incorporated into the PMMA, which will effect the mechanical properties obviously. This is a fatal issues in the scientific work.
(4) In the response, the author state “The concentration of graphene in our study has been analyzed in Raman spectra and has been quantified and confirmed by the phase rule as 0.027% by weight”, while in the revised manuscript, “RGO powder [0.1 (wt./wt.%)] was added to the MMA monomer”.
(5) In the revised manuscript, GO-PMMA composites was also prepared, while there is no mechanical properties for the GO-PMMA composites.
(6) As mentioned in the first round of review, why does the obtained composites discs is light yellow? The author replied “The colouring of the discs is adjusted with the use of other colouring additives that have no effect on the mechanical properties”. What is the colouring additives? Which does not mentioned in the materials section. In addition, colouring additives is always toxic compound, which is not suitable used in the dental prostheses. This is a fatal issues in the scientific work design.
Author Response
Dear reviewer:
Once again we thank you again for all your work in reviewing our article. Thank you for your time. Our contribution was focused on mechanical properties, especially friction and wear coefficients as well as mechanical properties in relation to composites used in dental prosthesis, which are original and which we think can contribute to the knowledge of this material for its dental applications.
Thanks to the reviewer's suggestions we have extended the study to the synthesis and characterization part and thus improved the paper.
We hope that all our efforts will serve to change the reviewer's opinion and we can count on his approval as the rest of the reviewers.
Best regards.
Comments and Suggestions for Authors
Though the author made a revision of this manuscript, yet more serious issues were appeared. The revised manuscript was also in bad organizing, , which is too primary to be a scientific paper at this stage. Therefore, a Rejection of this manuscript is recommended.
(1) The question “What is the Mn and PDI of the used PMMA?” present in the first round of review was not answered.
Done: According to the reviewer's request, both Mn and PDI of the PMMA used have been determined by using GPC techniques. These values have been introduced in the article.
Molecular weight (Mn) was determined by Gel Permeation Chromatography (GPC) using a liquid chromatografic pump (Shimadzu, model LC-8A, Tokyo, Japan) controlled by the LC Solution software (Shimadzu, Tokyo, Japan). The polymer was dissolved end eluted in 1,1,1,3,3,3-hexafluoroisopropanol (HFIP) containing CF3COONa (0.05M). The flow rate was 1 mL/min, the injected volume 20 mm and the sample concentration 6 mg/mL. A PL HFIPgel column (Agilent Technologies Deutschland GmbH, Boblingen, Germany) and a refractive index detector (Shimadzu, model RID-20A, Tokyo, Japan) were employed. The number and weight average molecular weights were determined using polymethylmethacrylate standards purchased from Sigma Aldrich.
(2) The question “What is the procedure for the preparation of composites discs?” present in the first round of review was also not answered.
DONE by explication:
In the last two decades, commercial companies in search of improving PMMA dental crowns have created and marketed resins in the form of stable solid blocks, because they have been previously polymerized at ideal temperatures and pressures, where minimal shrinkage occurs and a minimal amount of residual monomer is formed. This prior polymerization causes a decrease in the amount of internal porosity of the material, which is reflected in a higher final density of the material and a higher mechanical strength of the manufactured component. This type of block comes in different sizes, shapes and color ranges, although certain size standards are usually used, such as the one used in this study, with external diameters of between 98 and 99 mm and variable thicknesses of between 10 and 35 mm.
Computer-aided design/computer-aided manufacturing(CAD/CAM) techniques have expanded recently to embracethe fabrication of complete dentures, record bases, immediatedentures, and implant-supported overdentures in 2 clinical appointments.[23]. As CAD/CAM dentures are milled from pre-polymerized PMMA billets that are polymerized under high temperatures and pressure values, CAD/CAM dentures are reported to be less porous, and consequently, less likely to harbor virulent microorganisms such as Candida albicans, which will be less able to adhere to the surface of digitaldentures [24].
(3) As we all know the PMMA should be transparent and colorless, why does the PMMA obtained here is white color and opaque? Some white inorganic filler mus t be incorporated into the PMMA, which will effect the mechanical properties obviously. This is a fatal issues in the scientific work.
DONE: The following modification has been introduced in the text of the article in response to the reviewer's question.
The manufacturer of the raw material analyzed in this study offers the possibility of acquiring graphene-reinforced PMMA-based injected thermoplastic polymer discs, with dimensions of approximately 98 mm outside diameter with thicknesses between 12 and 30 mm, as well as with a chromatic range of multiple colors. However, in this study, a transparent base material without coloration but with maximum graphene loading has been analyzed in order to isolate the effect of graphene on the material properties avoiding any effect generated by the possible inorganic coloring fillers used for PMMA coloration.
A photographic image has been included in Figure 1 of one of the disks of material used in our characterization, where the semi-transparency of the material can be observed, with absence of coloration due to the absence of coloring additives but with a slight darkening caused only by the graphene content.
(4) In the response, the author state “The concentration of graphene in our study has been analyzed in Raman spectra and has been quantified and confirmed by the phase rule as 0.027% by weight”, while in the revised manuscript, “RGO powder [0.1 (wt./wt.%)] was added to the MMA monomer”.
DONE by justification:
We have to apologise for the error as the value is 0.1027 instead of 0.0027. A loss of that magnitude in the composite would not make sense. It is due to a typing error.
Although the manufacturer has indicated an approximate percentage of graphene reinforcement around 0.1(wt./wt.%), as a result of our characterization we have quantified a real percentage of reinforcement around 0.1027 wt./wt.%. The mismatch between the theoretical and real values of graphene reinforcement percentage could be explained in terms of heterogeneity in the distribution of the reinforcement phase inside the polymeric matrix.
(5) In the revised manuscript, GO-PMMA composites was also prepared, while there is no mechanical properties for the GO-PMMA composites.
DONE by justification:
In response to the reviewer's request for clarification, we should point out that new material based on PMMA plus graphene oxide has not been introduced, what we have pointed out is that a polymeric material based on PMMA reinforced with graphene oxide has been used. We regret the misunderstanding, which undoubtedly may be due to a bad wording of the text, but which we hope will be sufficiently clarified with this explanation. The method of preparation of PMMA and its reinforcement with GO has been described and introduced in the text.
(6) As mentioned in the first round of review, why does the obtained composites discs is light yellow? The author replied “The colouring of the discs is adjusted with the use of other colouring additives that have no effect on the mechanical properties”. What is the colouring additives? Which does not mentioned in the materials section. In addition, colouring additives is always toxic compound, which is not suitable used in the dental prostheses. This is a fatal issues in the scientific work design.
DONE by justification: Unfortunately, this misunderstanding is due to an error of ours in the writing of the article.
As we have explained before in section 3 of these reviews, the manufacturer of the material offers the possibility of acquiring this material with different chromatic ranges, adjusting the color by using unspecified inorganic additives (industrial secret). Despite this, in this study we have analyzed a transparent base material without coloration, in order to isolate the effect of graphene on the material properties avoiding any possible effect and/or interference of the inorganic coloring fillers.
However, we disagree with the reviewer regarding the toxicity of the material, as all current materials used in the dental field are subject to international Standard regulations such as ISO 10993 “Biological evaluation of medical devices “. The material marketed by the manufacturer has satisfactorily complied with a whole battery of tests described in the ISO 10993 standard mentioned above, such as the cell cytotoxicity tests described in the ISO-10993-5 standard "Biological evaluation of medical devices - Part 5: Tests for in vitro cytotoxicity" and the genotoxicity tests described in the ISO 10993-3 standard "Biological evaluation of medical devices - Part 3: Tests for genotoxicity, carcinogenicity and reproductive toxicity", as well as the bacterial mutation reversal tests described in OECD 471: 1997 "Guideline for testing of chemicals: Bacterial Reverse Mutation Test", as indicated on the manufacturer's website.
Submission Date
03 June 2022
Date of this review
05 Jul 2022 14:25:03
Reviewer 3 Report
The article level has been significantly improved. After a slight editorial mistake, it can be published. However, I am asking for a careful revision of the amendments made. I noticed citation errors in several places.
Author Response
Thank you very much for your comments. We have carried out a further review and have corrected some errors that we have detected.
Thank you very much for your time and help.
Best regards